# Effects of Drying Methods on the Volatile Compounds of *Allium*
*mongolicum* Regel

**DOI:** 10.3390/foods11142080

**Published:** 2022-07-13

**Authors:** Ledao Zhang, Shiying Cao, Junfang Li, Guoze Wang

**Affiliations:** School of Life Science and Technology, Inner Mongolia University of Science and Technology, Baotou 014010, China; ldzhang@imust.edu.cn (L.Z.); csy15248060758@126.com (S.C.); lijunfang_2005@126.com (J.L.)

**Keywords:** *Allium mongolicum* Regel, volatile compounds, headspace gas chromatography mass spectrometry (HS-GC-MS), characteristic volatiles fingerprinting, principal component analysis (PCA), drying methods

## Abstract

*Allium mongolicum* Regel (AMR) is a traditional Mongolian food. Various drying methods play an important role in foodstuff flavor. However, the effect of different drying methods on AMR is limited. In this study, freeze drying (FD), vacuum drying (VD), and hot-air drying (HAD) were applied to dry fresh AMR to a moisture content of 8% (wet basis); headspace gas chromatography mass spectrometry was adopted to identify volatile compounds in AMR; and principal component analysis and fingerprint similarity analysis based on the Euclidean distance was used to distinguish the fresh and three dried treatments. In total, 113 peaks were detected and 102 volatile compounds were identified. Drying causes significant changes to the amounts of volatile compounds in AMR, and the drying method plays a key role in determining which volatile compounds appear. Compared to FD, VD and HAD were more appropriate for drying AMR because the volatile compounds after VD and HAD were closer to those of fresh AMR. These findings can provide a scientific basis to help to preserve future seasonal functional food and aid in Mongolian medicine production.

## 1. Introduction

*Allium mongolicum* Regel (AMR), a perennial Liliaceae herb native to Mongolia, Kazakhstan, and China (Inner Mongolia, Gansu, Ningxia, and Xinjiang) [1] and also known as the Mongolia leek, grows in high-altitude desert steppe and desert areas [2]. AMR is resistant to wind erosion, drought, and low temperature and is therefore important in preventing wind erosion as well as in maintaining and improving regional ecological environments.

AMR is a traditional Mongolian medicinal herb [3]. Its leaves are rich in nutrients, possess a unique flavor, and is a natural green uncontaminated food [4] used to make sauces in cold vegetable dishes and stuffing of steamed stuffed buns and dumplings. In addition, AMR is beneficial in managing functional constipation, helping to maintain colon water content and increasing intestinal transit [1]. The aqueous extract of AMR has the potential to be used as a functional food or nutraceutical in the prevention and treatment of obesity and hypertension [3].

AMR is widely distributed on grassland, and it is eaten by both humans and sheep. It is found that sheep meat from AMR-pasture-fed animals tastes better. Zhao et al. [5] found that feeding AMR to sheep could significantly improve the quality of mutton.

Some research yielded AMR extracts to feed sheep [2,6]. The water-insoluble flavonoids from AMR were found to reduce the unpleasant mutton flavor and improve meat quality of sheep. AMR polysaccharide supplementation significantly increased serum total protein, albumin, and globulin concentrations and reduced mortality after LPS challenge [7]. AMR supplementation can also affect nutrient digestibility, CH_4_ emission, and the antioxidant capacity of Simmental calves in northwest China [8].

However, fresh AMR is seasonal and perishable because of its high moisture content, making AMR difficult to be enjoyed year round. Overcoming this seasonality is crucial to utilizing AMR. Drying, as an essential processing operation, can reduce moisture to a safe level at which microorganisms and deterioration reactions are inhibited. Freeze drying (FD) has the highest cost, but produces the highest quality food products among all drying methods [9,10]. Hot-air drying (HAD) is low-cost and the most common method that is applied to the storage of agricultural products [11,12]. The main drawbacks of HAD are undesirable physical, structural, chemical, organoleptic, and nutritional changes [13]. Vacuum drying (VD) with low pressure and low drying temperatures can yield a higher quality than atmospheric drying [14].

Flavor is mainly determined by volatile components and is a key factor for consumers to accept dried AMR. Headspace gas chromatography mass spectrometry (HS-GC-MS) is a new and reliable method used to identify volatile compounds in foods, such as fermented rose jams [15], Dezhou braised chicken [16], dry-cured fish [17], yak milk [18], soybeans [19], tea [20], Chinese prickly ash peels [21], canned bamboo shoots [22], and fresh-cut yams [23]. HS-GC-MS was used to analyze the changes of volatile components in fried *Tricholoma matsutake* Singer under different heating temperatures and times [24], and a total of 40 signals that corresponded to 24 compounds were identified. The effects of spray drying and FD on volatile components of yak milk powder were compared by Feng et al. [18]. Their results indicated that different drying methods affect the flavor of yak milk powder. Volatile components in Acacia honey powders were found to have significant differences under various drying conditions [25]. Changes in the volatile compounds of peppers during the drying process based on HS-GC-MS was studied by Ge et al. [26]. Their results indicated that the change of volatile compounds is significantly determined by the drying temperature.

Once moisture is removed from AMR, it has a potential to be used as a feed supplement and edible spice over a long term. However, dried AMR still has not been widely adopted. Until now, the kinds of drying methods that can be employed to dry AMR and retain flavor to its maximum are still not subject to research. In this study, HS-GC-MS was used to detect volatile components of AMR after processing by HAD, VD, and FD. The effect of these drying methods on the volatile compounds was studied with principal component analysis (PCA) and Euclidean distance analysis. Finally, the knowledge obtained from this study has the potential to be used to improve the industrial processing of AMR traditional feed supplements and edible production.

## 2. Materials and Methods

### 2.1. Materials

Fresh aerial parts of AMR were purchased from a local market in Baotou, Inner Mongolia, China, in August 2021. These parts were washed and the surface water of the sample was removed by centrifugation. Then, samples were cut into segments of 5 cm in length. Portions of each sample were immediately stored at 4 °C in sealed plastic bags as samples. Before the drying experiment began, the AMR in a sealed plastic bag was taken out and placed indoors until its temperature stabilized. The other samples were placed in a polypropylene tray and frozen at −25 °C for at least 8 h as the frozen samples. The initial moisture content of the sample was 91.93% (wet basis).

### 2.2. Drying Experiments

The AMR samples were dried by using different drying methods until the final moisture content was <8% (wet basis). These methods are described below. After drying, samples were vacuum-packed and stored for 10 days at ambient temperature (25 °C ± 5 °C) for subsequent analyses.

#### 2.2.1. FD Process

The frozen samples (500 g) were dried by using a freeze dryer (SCIENTZ-18N, Ningbo Xinzhi Biotechnology Co., Ltd., Ningbo, China). The cold trap temperature was set at −60 °C, and the pressure of the drying chamber was set at 2 Pa during the drying process. Drying was completed after 29 h.

#### 2.2.2. VD Process

The samples (500 g) were dried by using a vacuum dryer (DZF-6050B, Shanghai Yiheng Science Co., Ltd., Shanghai, China). The drying temperature was set at 50 °C and the pressure of the drying chamber was set at 2.0 × 10^4^ Pa during the drying process. Drying was completed after 36 h.

#### 2.2.3. HAD Process

The samples (500 g) were dried by using a hot-air dryer (ZXRD-B5110, Shanghai Zhicheng Analytical Instrument Manufacturing Co., Ltd., Shanghai, China). The drying temperature was set at 50 °C. Drying was completed after 46 h.

### 2.3. HS-GC-MS Analysis

The analyses were performed on a GC-MS (FlavourSpec^®^, Gesellschaft für Analytische Sensorsysteme mbH, Dortmund, Germany) equipped with an autosampler (CTC Analytics AG, Zwingen, Switzerland) with a headspace sampling unit and a 1 mL gas-tight syringe (Gerstel GmbH, Mühlheim, Germany). The GC was equipped with an MXT-WAX capillary column (30 m × 0.53 mm × 1 μm).

The column temperature was 60 °C, and the temperature of the MS was 45 °C. Nitrogen of 99.99% purity was used as a carrier gas and drift gas. The carrier gas was programmed to flow as follows: 2 mL/min during 0–2 min, 2–10 mL/min during 2–10 min, 10–100 mL/min during 10–20 min, and 100–150 mL/min during 20–40 min. The drift gas was set at 150 mL/min.

### 2.4. Statistical Analysis

Data from volatile compounds in samples were acquired and processed using Laboratory Analytical Viewer analysis software and Library Search qualitative software (G.A.S., Dortmund, Germany). Laboratory Analytical Viewer was used to view the analytical spectrum, where each dot represents a volatile compound. A reporter plugin was directly used to compare the spectral differences between samples (two-dimensional and three-dimensional views). A gallery plot plugin was used to compare fingerprints and visually and quantitatively compare the differences in volatile components among different samples. A dynamic PCA plugin was used for dynamic PCA and clustering analysis of the samples and to quickly determine the types of unknown samples. GC-IMS Library Search is an application software that was used to qualitatively analyze substances based on the NIST2014 Mass Spectral Library and IMS Library.

## 3. Results and Discussion

### 3.1. Visual Topographic Plot Comparison

The differences in volatile compounds in AMR samples from fresh and different drying methods were analyzed by using HS–GC–IMS. The data are represented by three-dimensional topographical visualizations in Figure 1a, where the *Y*-axis represents the retention time of the gas chromatograph, the *X*-axis represents the ion migration time for identification, and the *Z*-axis represents the peak height for quantification. As can be seen from Figure 1a, the volatile compounds and the signal intensity of AMR of fresh and dried samples from different drying methods are significantly different. The volatile compounds in fresh AMR have maximum intensities. Compared with the fresh samples, some volatile compounds were generated and some volatile components disappeared, so the contents of some volatile components decreased and the contents of other volatile components increased in dried samples. This phenomenon was also observed by Guo et al. [24]. They reported that some compounds dramatically decreased and some volatile compounds were formed once processed by HAD.

The data are represented by two-dimensional topographical visualizations in Figure 1b. In Figure 1b, the ion migration time and the position of the reactive ion peak were normalized. The fingerprint shows the total volatile components of the AMR samples. Each dot on the fingerprint represents a single volatile compound separated from the total volatile components. The color of the dots represents the signal intensity of the volatile components. The redder the color, the greater the signal intensity and the higher the content of the target volatile compound. As can be clearly seen from Figure 1b, the fingerprint of fresh AMR is obviously different from that of the dried AMR samples, with most of the signals within a retention time of 30–1000 s and with a drift time of 1.0–1.75. The fingerprints of the three dried AMR samples were similar to each other, but their signal intensities were slightly different. This means that each dried AMR sample had a unique specific flavor.

The difference comparison model was applied to compare the differences of AMR samples. The topographic plot of fresh AMR was selected as a reference, and the topographic plot of dried samples was deducted from the reference. The results are shown in Figure 2. The white, red, and blue colors in dried samples indicate that the concentration of the volatile compounds is consistent with that of the reference, higher than that of the reference, and lower than that of the reference, respectively. It can be seen that most of the signals in the topographic plot of fresh and dried samples from different areas appear within a retention time of 30–1300 s. After drying, the signal intensities of some compounds decrease, while others disappear completely. In contrast, some signal intensities increase, indicating that the content of some compounds increases after drying. It also can be seen from Figure 2 that the signals for a retention time between 600 and 700 s in FD are intense, but in VD and HAD disappear. The different drying methods lead to variations in the volatile compounds in the dried AMR samples.

### 3.2. Effects of the Different Drying Methods on the Volatile Compounds in the AMR Samples

To further compare the specific volatile compounds in each group of samples, all peaks were selected for fingerprint plot comparison, as shown in Figure 3. Columns represent the detected substance, and rows represent the content of the same volatile compounds in different samples. Individual dots represent a volatile compound, and the color represents the content levels of the volatile compounds (the redder and brighter the color, the higher the content). In the fingerprint plot, unidentified volatile compounds are represented by numbers, and some volatile compounds with monomer and dimer morphologies were detected.

The analysis of this fingerprint plot clearly illustrates the differences in volatile components of AMR with different drying methods. As shown in Figure 3, 2-methylpropy-l-butanoate, isopulegol, dihydro-5-methyl-2(3H)-furanone, dimethyl trisulfide, ethyl heptanoate, propyl disulfide, gamma-butyrolactone, methyl propyl trisulfide M, methyl benzoate, ethyl propanoate, 2,3-dimethylpyrazine M, diallyl sulfide, 2-methyl-3-furanthiol, propyl butyrate M, pentanoic acid, butyl 2-methylbutanoate, 2,3-dimethylpyrazine D, propyl 1-propenyl disulfide D, methyl salicylate, ethyl 2-hydroxy-4-methylpentanoate, trimethyl pyrazine, methyl propyl trisulfide D, dipropyl trisulfide D, meta-cresol, heptanol, 3-acetyl-6-methyl-2H-pyran-2,4(3H)-dione M, 3-acetyl-6-methyl-2H-pyran-2,4(3H)-dione D, 2-hydroxy-3-methyl-2-cyclopentene-1-one (cyclotene), octen-3-ol, and butyl hexanoate, labelled as A, were detected in fresh AMR samples and were non-existent in dried AMR samples, so these compounds can be extremely damaged after drying.

2-Propanone, isopentanol, 2-butanone, butanal, methyl acetate, 1-penten-3-ol, isobutyric acid, ethyl trans-2-butenoate, 2-butanol, 3-methyl-2-butenal, hexanal, 1-penten-3-one, heptanal, 6-methyl-5-hepten-2-one, 3-methyl butanal D, 2-methyl-1-heptene, and 2,6-dimethyl pyrazine, labelled as B, were not detected in fresh AMR samples, but exist in dried AMR samples, so there were generated during drying. Alpha-terpinene M, 1,8-cineole M, isovaleric acid, alpha-phellandrene, 2,3-dihydro-4-hydroxy-2,5-dimethyl-3-furanone, hexane nitrile, 2,3-dimethyl-5-ethyl pyrazine, n-hexanol, 1,8-cineole D, methyl hexanoate, alpha-terpinene D, (Z)-4-heptenal, and 2-methyl-2-pentenal D, labelled as C, were not detected in fresh AMR samples, but only exist in the FD AMR samples, so these compounds can only be formed in FD.

The amounts of dimethyl disulfide, methyl propyl disulfide M, methyl propyl disulfide D, amyl acetate, ethyl hexanoate M, dipropyl trisulfide M, ethyl hexanoate D, propyl butyrate D, 2-ethyl furan, 1-butanol, (3E)-hexenol, (E)-2-hexenal, 3-hepten-2-one, and 2-methyl-1-butanol, labelled as D, dramatically decreased once the fresh AMR samples were dried. The amounts of 3-methyl butanal M, 2-hexen-1-ol M, and 2-hexen-1-ol D, labelled as E, were similar in both the fresh and dried AMR samples, so these compounds cannot be significantly damaged by the drying methods used in this study.

The amounts of methional, N-nitroso-morpholine, propyl 1-propenyl disulfide M, 2,3-diethyl-5-methyl pyrazine, propyl hexanoate, 2-acetyl furan and diallyl disulfide, labelled as E, decreased in dried AMR samples and their signal intensity was similar among different drying methods. Ethyl sulfide and ethyl acetate, labelled as E, suffered the most serious damage under FD and suffered less damage under VD and HAD. The amount of 2-methyl-2-pentenal M, labelled as E, was not decreased in FD, but almost vanished in VD and HAD, that is, 2-methyl-2-pentenal M in fresh AMR samples can only be preserved well by FD. This means that 2-methyl-2-pentenal M can be protected well by low temperature and sublimation.

Volatile components labelled as F can appear after drying. Of these, hexyl butanoate formed in HAD; acetoin formed in VD and HAD; 3-methyl-2-pentanone, methyl 3-methyl butanoate, and 3-methyl-1-pentanol formed in VD; and 2-ethyl-1-hexanol formed in FD and VD.

### 3.3. Compound Identification

After analyzing AMR samples under different drying methods, we tentatively identified a total of 113 typical target signals by comparing the feature retention and drift times with those of the individual standard ion signals. These were confirmed by using the commercial GC-IMS Library Search and are represented by different numbers (1–113). The identified and unidentified compounds are listed in Table 1. The identified compounds include the compound name, CAS number, molecular formula, molecular weight, retention index, retention time, and drift time. The unidentified compounds include the retention index, retention time, and drift time. The tentatively identified volatile compounds in AMR samples included 5 ketones, 9 aldehydes, 12 alcohols, 9 esters, 4 acids, 3 ethers, 5 alkenes, 8 salts, 2 furans, 6 pyrazines, and 10 sulfide compounds.

### 3.4. Cluster Analysis of the Fresh and Dried AMR Samples

PCA is a multivariate statistical analysis technique established using signal intensity to highlight the differences in volatile compounds. The PCA results of the volatile compounds in the fresh and dried AMR samples are presented in Figure 4. The figure shows the distribution map for the first two principal components determined by PCA, which describe 69% and 18%, respectively, of the accumulative variance contribution rate, and a visualization of the data was obtained. These components were thought to show a similarity between the different AMR samples.

The PCA results clearly show fresh and different dried AMR samples in a relatively independent space and these would be well distinguished in the distribution map. Dried AMR samples can be well distinguished according to the positive score values of PC1, while fresh samples can be well defined according to the negative scores of PC1, and the difference in different parts from different areas can be separated by combining the scores with the score values of PC2.

From Figure 4, one can see that flavors varied greatly between the fresh and dried AMR samples. Moreover, the flavors of FD AMR samples were vastly different from the flavors of VD and HAD AMR samples. The flavors of VD and HAD AMR samples were relatively similar. The results revealed that the characteristic volatile fingerprints of the fresh and dried AMR samples under different drying methods were successfully established through HS-GC–IMS. The HS-GC–IMS data contained valuable information and can be a useful tool for distinguishing AMR samples. Feng et al. [18] also reported that volatile compounds in yak milk powder have significant differences between FD and spray drying.

Figure 5 shows the fingerprint similarity based on Euclidean distance. The Euclidean distance between AMR samples was 38,613–30,496,849. The mean values of FR and FD samples differed by 27,806,152; the mean values of FR and VD samples differed by 24,608,500; and the mean values of FR and HAD samples differed by 21,211,236. When Euclidean distance was larger, the distance between samples was father apart, and the similarity differences in fingerprint spectra became obvious. Thus, compared to VD and HAD, the difference between the FD and fresh AMR samples was more significant. The reason for this is the amount of volatile components that formed after FD, as shown in the C part of Figure 3. Some reactions produced or eliminated volatile components under high-vacuum, low-temperature conditions.

## 4. Conclusions

In this study, a total of 113 signal peaks from topographic plots were identified in the fresh and dried AMR samples treated by using different drying methods. The drying process changes the amounts of volatile components found in the fresh AMR. Different drying methods significantly affected the volatile components of AMR. A total of 28 new volatile components formed under the operation of FD, VD, and HAD, of which 14 new volatile components can only be formed in FD. Thus, the differences of volatile components between FD samples and fresh samples were much greater than those between fresh samples and VD and HAD samples. That is, FD is unsuitable for attaining dried AMR samples if the maximum amounts of volatile components in dried samples is required. In this study, the volatile component variation under FD, VD, and HAD was discussed in detail, and the basic experimental data were supplied for different requirements based on the choice of volatile components desired.

## Figures and Tables

**Figure 1 foods-11-02080-f001:**
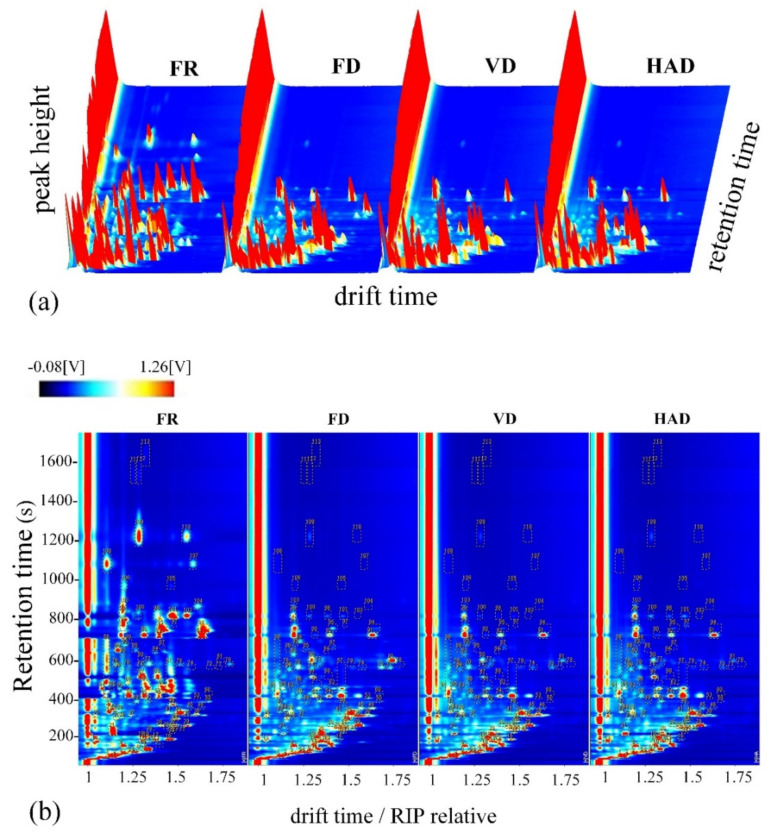
Topographic plots of fresh (FR) and dried AMR treated by using different drying methods. (**a**) Three-dimensional topographic plots. (**b**) Two-dimensional topographic plots.

**Figure 2 foods-11-02080-f002:**
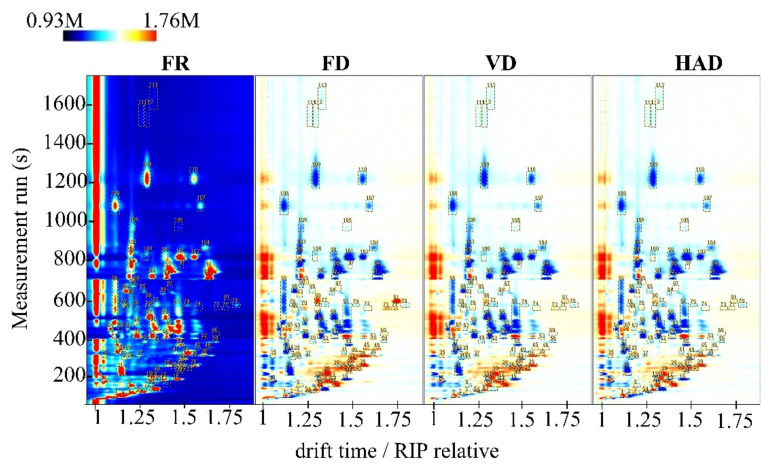
Differential HS-GC-MS (two-dimensional topographic) images of fresh and dried AMR treated by using different drying methods.

**Figure 3 foods-11-02080-f003:**
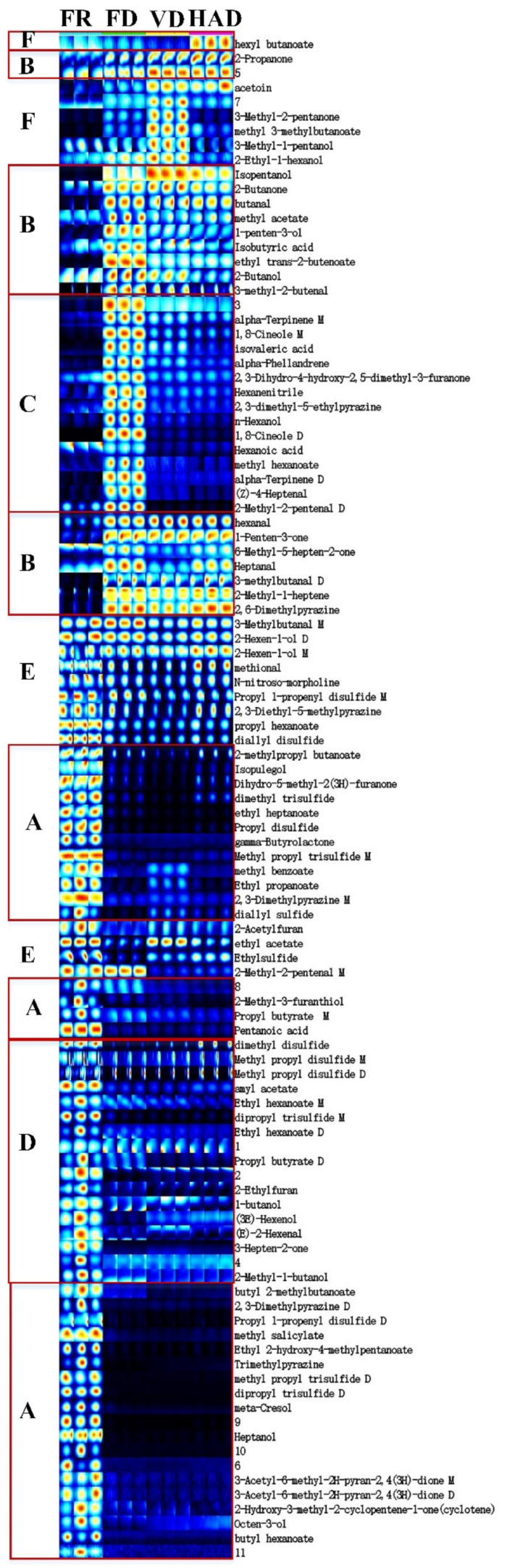
Fingerprint plot of volatiles of fresh and dried AMR treated by using different drying methods.

**Figure 4 foods-11-02080-f004:**
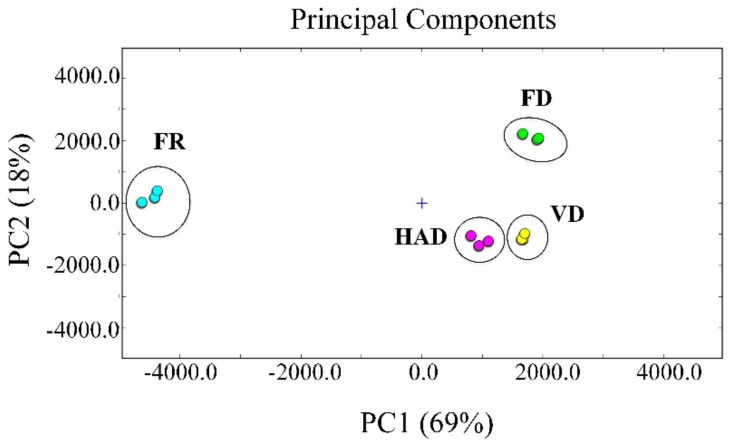
PCA analysis plot of the flavor of the fresh and dried AMR samples treated by using different drying methods.

**Figure 5 foods-11-02080-f005:**
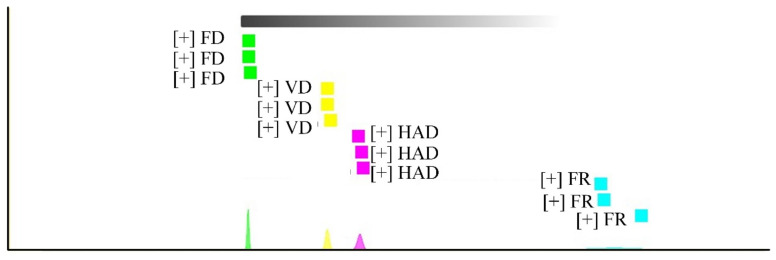
Fingerprint similarity based on the Euclidean distance of the fresh and dried AMR samples treated by using different drying methods. (green indicated FD, yellow indicated VD, pink indicated HAD, blue indicated FR).

**Table 1 foods-11-02080-t001:** GC-MS integration parameters of the fresh and dried AMR samples.

Count	Compound	CAS	Formula	MW	RI	Rt [s]	Dt [a.u.]
1	2-Propanone	C67641	C_3_H_6_O	58.1	536.3	108.446	1.12414
2	2-Butanol	C78922	C_4_H_10_O	74.1	548.5	113.572	1.15347
3	Methyl acetate	C79209	C_3_H_6_O_2_	74.1	560.8	118.706	1.19142
4	1	unidentified	*	0	572.6	123.67	1.2584
5	2-Butanone	C78933	C_4_H_8_O	72.1	584.7	128.747	1.24163
6	Butanal	C123728	C_4_H_8_O	72.1	598.9	134.743	1.28504
7	Ethyl acetate	C141786	C_4_H_8_O_2_	88.1	607.3	138.248	1.33671
8	3-Methyl butanal M	C590863	C_5_H_10_O	86.1	651.5	156.814	1.19727
9	3-Methyl butanal D	C590863	C_5_H_10_O	86.1	653.7	157.754	1.39903
10	1-Penten-3-one	C1629589	C_5_H_8_O	84.1	652.1	157.046	1.31
12	1-Penten-3-ol	C616251	C_5_H_10_O	86.1	696.1	178.41	1.35294
13	Ethyl sulphide	C352932	C_4_H_10_S	90.2	697.2	179.314	1.04603
14	2-Ethyl furan	C3208160	C_6_H_8_O	96.1	699.5	181.214	1.30444
15	1-Butanol	C71363	C_4_H_10_O	74.1	703.9	184.755	1.37849
16	Ethyl propanoate	C105373	C_5_H_10_O_2_	102.1	707.8	187.894	1.45024
17	Isobutyric acid	C79312	C_4_H_8_O_2_	88.1	719.6	197.349	1.37582
18	Acetoin	C513860	C_4_H_8_O_2_	88.1	723.5	200.563	1.32893
19	2	unidentified	*	0	728.8	204.801	1.29868
20	3-Methyl-2-butenal	C107868	C_5_H_8_O	84.1	741.8	215.293	1.34865
21	Dimethyl disulphide	C624920	C_2_H_6_S_2_	94.2	743.3	216.498	1.13845
22	3-Methyl-2-pentanone	C565617	C_6_H_12_O	100.2	752.8	224.139	1.47277
23	3	unidentified	*	0	768.9	237.157	1.45924
24	Methyl 3-methyl butanoate	C556241	C_6_H_12_O_2_	116.2	769.5	237.621	1.53098
25	Isopentanol	C123513	C_5_H_12_O	88.1	770.2	238.197	1.4841
26	2-Methyl-1-heptene	C15870107	C_8_H_16_	112.2	782.6	248.142	1.47952
27	2-Methyl-1-butanol	C137326	C_5_H_12_O	88.1	792.2	258.085	1.48073
28	4	unidentified	*	0	792.5	258.384	1.40941
29	Hexanal	C66251	C_6_H_12_O	100.2	793.6	259.596	1.55782
30	Isovaleric acid	C503742	C_5_H_10_O_2_	102.1	813	280.561	1.4854
31	2-Methyl-2-pentenal M	C623369	C_6_H_10_O	98.1	826.8	295.433	1.15696
32	2-Methyl-2-pentenal D	C623369	C_6_H_10_O	98.1	825.2	293.713	1.49264
34	(E)-2-Hexenal	C6728263	C_6_H_10_O	98.1	830.7	299.661	1.18611
35	5	unidentified	*	0	838.9	308.472	1.04637
36	Ethyl trans-2-butenoate	C623701	C_6_H_10_O_2_	114.1	838.9	308.472	1.55657
37	(3E)-Hexenol	C928972	C_6_H_12_O	100.2	841.8	311.6	1.24037
38	3-Methyl-1-pentanol	C589355	C_6_H_14_O	102.2	845.2	315.253	1.60253
39	2-Hexen-1-ol M	C2305217	C_6_H_12_O	100.2	854.1	324.836	1.18102
40	2-Hexen-1-ol D	C2305217	C_6_H_12_O	100.2	852.3	322.94	1.51239
41	2-Acetylfuran	C1192627	C_6_H_6_O_2_	110.1	853.3	324.055	1.44499
43	Diallyl sulphide	C592881	C_6_H_10_S	114.2	856.8	327.794	1.11914
44	2-Methyl-3-furanthiol	C28588741	C_5_H_6_OS	114.2	870.4	342.406	1.13906
45	Hexanenitrile	C628739	C_6_H_11_N	97.2	873.3	345.59	1.56901
46	n-Hexanol	C111273	C_6_H_14_O	102.2	874.8	347.211	1.63282
47	2,3-Dimethyl pyrazine M	C5910894	C_6_H_8_N_2_	108.1	893.6	368.454	1.11623
48	2,3-Dimethyl pyrazine D	C5910894	C_6_H_8_N_2_	108.1	892.1	365.857	1.47291
49	Propyl butyrate M	C105668	C_7_H_14_O_2_	130.2	901	381.438	1.27041
50	Propyl butyrate D	C105668	C_7_H_14_O_2_	130.2	900.1	379.934	1.67426
52	Heptanal	C111717	C_7_H_14_O	114.2	902.1	383.523	1.34098
53	(Z)-4-Heptenal	C6728310	C_7_H_12_O	112.2	910.6	398.436	1.61085
54	Methional	C3268493	C_4_H_8_OS	104.2	916.4	408.811	1.39433
55	Amyl acetate	C628637	C_7_H_14_O_2_	130.2	917.4	410.432	1.3096
56	Methyl propyl disulphide M	C2179604	C_4_H_10_S_2_	122.2	918.7	412.813	1.10294
57	Methyl propyl disulphide D	C2179604	C_4_H_10_S_2_	122.2	920.1	415.249	1.4606
58	Pentanoic acid	C109524	C_5_H_10_O_2_	102.1	919.1	413.431	1.22958
60	Methyl hexanoate	C106707	C_7_H_14_O_2_	130.2	923.8	421.779	1.67884
61	2-Methyl propyl butanoate	C539902	C_8_H_16_O_2_	144.2	938	446.968	1.33013
62	Dihydro-5-methyl-2(3H)-furanone	C108292	C_5_H_8_O_2_	100.1	938.4	447.656	1.41902
63	6	unidentified	*	0	938.9	448.447	1.17791
64	Gamma-butyro lactone	C96480	C_4_H_6_O_2_	86.1	939	448.742	1.08198
65	2,6-Dimethyl pyrazine	C108509	C_6_H_8_N_2_	108.1	939.9	450.31	1.13806
66	3-Hepten-2-one	C1119444	C_7_H_12_O	112.2	954.9	476.756	1.22602
67	Dimethyl trisulphide	C3658808	C_2_H_6_S_3_	126.3	972.5	507.975	1.30465
68	Heptanol	C53535334	C_7_H_16_O	116.2	973.2	509.037	1.38823
69	6-Methyl-5-hepten-2-one	C110930	C_8_H_14_O	126.2	993.3	544.732	1.17459
70	Alpha-terpinene M	C99865	C_10_H_16_	136.2	1000.2	557.635	1.21973
71	Alpha-terpinene D	C99865	C_10_H_16_	136.2	997.6	552.535	1.72939
72	Hexanoic acid	C142621	C_6_H_12_O_2_	116.2	997.8	552.907	1.29923
73	Alpha-phellandrene	C99832	C_10_H_16_	136.2	998.2	553.65	1.68527
74	7	unidentified	*	0	998.4	554.203	1.57452
76	Octen-3-ol	C3391864	C_8_H_16_O	128.2	1006.3	569.832	1.1572
77	Ethyl hexanoate M	C123660	C_8_H_16_O_2_	144.2	1007	571.264	1.34206
78	Ethyl hexanoate D	C123660	C_8_H_16_O_2_	144.2	1007.2	571.694	1.7949
79	2-Hydroxy-3-methyl-2-cyclopentene-1-one(cyclotene)	C80717	C_6_H_8_O_2_	112.1	1007	571.28	1.51506
80	1,8-Cineole M	C470826	C_10_H_18_O	154.3	1016.8	590.853	1.29735
81	1,8-Cineole D	C470826	C_10_H_18_O	154.3	1016.5	590.4	1.74455
83	2,3-Dihydro-4-hydroxy-2,5-dimethyl-3-furanone	C3658773	C_6_H_8_O_3_	128.1	1031.7	620.69	1.19636
84	Butyl 2-methyl butanoate	C15706737	C_9_H_18_O_2_	158.2	1036.6	630.593	1.3765
85	8	unidentified	*	0	1042.4	642.227	1.27137
86	Trimethyl pyrazine	C14667551	C_7_H_10_N_2_	122.2	1043.3	643.886	1.16751
87	2-Ethyl-1-hexanol	C104767	C_8_H_18_O	130.2	1049.7	656.694	1.41491
88	Methyl benzoate	C93583	C_8_H_8_O_2_	136.1	1055.4	668.249	1.20613
89	2,3-Dimethyl-5-ethylpyrazine	C15707343	C_8_H_12_N_2_	136.2	1064.8	686.901	1.23042
90	Ethyl 2-hydroxy-4-methyl pentanoate	C10348477	C_8_H_16_O_3_	160.2	1080.3	717.964	1.31203
91	Diallyl disulphide	C2179579	C_6_H_10_S_2_	146.3	1080.4	718.281	1.63547
92	Propyl hexanoate	C626777	C_9_H_18_O_2_	158.2	1081.1	719.699	1.3934
93	Propyl 1-propenyl disulphide M	C5905464	C_6_H_12_S_2_	148.3	1098.5	754.348	1.19864
94	Propyl 1-propenyl disulphide D	C5905464	C_6_H_12_S_2_	148.3	1095.9	749.146	1.64082
95	Ethyl heptanoate	C106309	C_9_H_18_O_2_	158.2	1097.8	753.049	1.40951
97	Hexyl butanoate	C2639636	C_10_H_20_O_2_	172.3	1106.5	770.488	1.48052
98	Isopulegol	C89792	C_10_H_18_O	154.3	1129.3	816.153	1.3872
99	N-Nitroso-morpholine	C59892	C_4_H_8_N_2_O_2_	116.1	1129.4	816.182	1.19266
100	2,3-Diethyl-5-methyl pyrazine	C18138040	C_9_H_14_N_2_	150.2	1129.8	817.092	1.28369
101	Propyl disulphide	C629196	C_6_H_14_S_2_	150.3	1130.6	818.6	1.46879
102	9	unidentified	*	0	1130.7	818.918	1.5482
103	Methyl propyl trisulphide M	C17619362	C_4_H_10_S_3_	154.3	1154	865.496	1.1932
104	Methyl propyl trisulphide D	C17619362	C_4_H_10_S_3_	154.3	1155.3	868.207	1.61442
105	Butyl hexanoate	C626824	C_10_H_20_O_2_	172.3	1207	971.657	1.46572
106	Methyl salicylate	C119368	C_8_H_8_O_3_	152.1	1208	973.543	1.20271
107	10	unidentified	*	0	1263.5	1084.592	1.59167
108	Meta-Cresol	C108394	C_7_H_8_O	108.1	1264	1085.714	1.10663
109	Dipropyl trisulphide M	C6028611	C_6_H_14_S_3_	182.4	1331.9	1221.641	1.28633
110	Dipropyl trisulphide D	C6028611	C_6_H_14_S_3_	182.4	1333.1	1223.954	1.55637
111	3-Acetyl-6-methyl-2H-pyran-2,4(3H)-dione M	C520456	C_8_H_8_O_4_	168.1	1491.9	1541.69	1.25381
112	3-Acetyl-6-methyl-2H-pyran-2,4(3H)-dione D	C520456	C_8_H_8_O_4_	168.1	1492.7	1543.284	1.28589
113	11	unidentified	*	0	1539.6	1637.292	1.31797

Notes: MW: molecular mass. RI: retention index. Rt [s]: retention time. Dt [a.u.]: the drift time. symbol* indicated the unidentified compounds.

## Data Availability

The date are available from the corresponding author.

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
