# Peer review of "Effects of Drying Methods on the Volatile Compounds of *Allium"

_foods, 2022, doi:10.3390/foods11142080_

Round 1

Reviewer 1 Report

Overall:

Title, abstract, key words and manuscript: Allium mongolicum

Please pay attention that a scientific species name comprises two words, both in italic, the first word starting with a capital letter but not the second: Allium mongolicum 

Thus, every time a scientific name is placed authors must accomplished this rule. ex: Tricholoma matsutake

As it is a traditional Mongolian vegetable it is important to name it. Consider using A. mongolicum (AM) instead of AMR (the name of its author Regel should appear just in the first mention to the species and there is no need to keep it along the manuscript). 

The paper shows innovation with respect to the use of drying methods to overcome the seasonal availability of A. mongolicum, including the choice of the best  drying method to retain their aroma profile  to the maximum extent.

The information placed (methods and results presentation) are correct, but the English is so poor that difficult the reading and the interpretation. I advise an editing service.

Author Response

Dear Reviewers:

Thank you for your comments concerning our manuscript entitled “The Effects of Drying Methods on Volatile Compounds of Allium mongolicum Regel” (ID: 1749155). The comments are very valuable and helpful for improving our manuscript. We have made modification carefully according to your comments. The revised parts are marked in red. The reply for your comments is in the following:

Reply to the reviewer 1 comments:

Reviever#1:

(1) Title, abstract, key words and manuscript: Allium mongolicum

Reply:

We have check and revised the “Allium mongolicum” in title, abstract, key words and manuscript, please see the correction in revised version.

(2) Please pay attention that a scientific species name comprises two words, both in italic, the first word starting with a capital letter but not the second: Allium mongolicum .Thus, every time a scientific name is placed authors must accomplished this rule. ex: Tricholoma matsutake

Reply:

We have changed “Allium Mongolicum” to “Allium mongolicum”, please see the correction in revised version.

(3) As it is a traditional Mongolian vegetable it is important to name it. Consider using A. mongolicum (AM) instead of AMR (the name of its author Regel should appear just in the first mention to the species and there is no need to keep it along the manuscript). 

Reply:

We have using “AM” instead of “AMR”, please see the correction in revised version.

(4) The paper shows innovation with respect to the use of drying methods to overcome the seasonal availability of A. mongolicum, including the choice of the best drying method to retain their aroma profile to the maximum extent.The information placed (methods and results presentation) are correct, but the English is so poor that difficult the reading and the interpretation. I advise an editing service.

Reply:

More English writing has been examined and polished through an editing service, please see the revised version.

Thank you very much for your help again. Please contact us freely if you have questions.

With best regards.

Sincrcely yours,

Zhang Ledao and Wang Guoze 

Reviewer 2 Report

This paper presents the effect of drying metod (hot air drying, vacuum drying, freeze drying) on volatile compounds of Allium Mongolicum Regel (AMR). The volatile profiles AMR change during drying with different methods. Headspace-gas chromatography-ion mobility spectrometry (HS-GC-IMS) is a promising technology for the accurate characterization and detection of volatile compounds in food products. The article is interesting, innovative, but has a few weak points.

No discussion of the results with the literature.

Text should be carefully formatted many words are joined together (no spaces).

English style editing required.

Other comments:

Allium Mongolium - Latin names should be written in italics, corrected in the title and text of the work.

L41. Explain whether it is about the smell or the quality of the meat? List the literature cited.

L42-43. ‘How to overcome seasonality of AMR?’ Style.

L60-63. For each example of food, assign literature separately. Example: ‘….fermented rose jams [14], Dezhou braised chicken [15]…’

L64. Tricholoma matsutake – Latin names should be written in italics.

L67.  ‘…yak milk powder were compared by [17] .’ Add the name of the first author of the cited publication.

L73. The sentence ‘Drying is an essential methods to overcome seasonality and extend the storage period of AMR.’ is a repetition of a sentence L50-52.  Rephrase or remove.

L74-L76. ‘However, dried AMR still has not been adopted. Until now, what kind of drying methods can be employed to dry AMR and retain flavour to the maximum extent  possible is still a research gap.’ Use English style!

Is there a need to dry AMR for human nutrition? I think that AMR is mainly consumed by farm animals (sheep). So, is it not mowed and dried in the sun? I propose to rebuild this paragraph to focus on the effects of drying on the volatile compounds.

L80. Change the word ‘utilized’ to ‘used’.

L91. ‘The initial moisture content of the sample was 91.93% (w.b.)’. How was it measured?

L93-94. How was it determined that a sample had a moisture content of less than 8%?

L95. At ambient temperature? Enter a value.

L118-129. Statistical analysis. How many repetitions was HS-GC IMS analysis performed? How many repetitions of drying were made for each of method?

L97-107. Add drying time to each drying method (FD, VD, HAD).

L116. ‘…10-100mL/min at 0-20 min…’ Should be 10-20 min?

L138-141. Style. Rephrase.

L152. meant?

L177-179. ‘…the brighter the color, the higher the content.’ What about the red one?

L272-275. Where do these values come from? In Fig. 5, the axis names and scale are missing.

L285-287. ‘…increased, decreased and vanished.’ What  does it mean? Is there any trend according to the drying method?

Author Response

Dear Reviewers:

Thank you for your comments concerning our manuscript entitled “The Effects of Drying Methods on Volatile Compounds of Allium mongolicum Regel” (ID: 1749155). The comments are very valuable and helpful for improving our manuscript. We have made modification carefully according to your comments. The revised parts are marked in red. The reply for your comments is in the following:

Reply to the reviewer 2 comments:

Reviever#2: This paper presents the effect of drying metod (hot air drying, vacuum drying, freeze drying) on volatile compounds of Allium Mongolicum Regel (AMR). The volatile profiles AMR change during drying with different methods. Headspace-gas chromatography-ion mobility spectrometry (HS-GC-IMS) is a promising technology for the accurate characterization and detection of volatile compounds in food products. The article is interesting, innovative, but has a few weak points.

(1) No discussion of the results with the literature.

Reply:

We have added the discussion of the results with the literature manuscript, please see the correction in revised version.

(2) Text should be carefully formatted many words are joined together (no spaces).

Reply:

We have added the discussion of the results with the literature manuscript, please see the L144-146 and L273-274 in revised version.

(3) Allium Mongolium - Latin names should be written in italics, corrected in the title and text of the work.

Reply:

We have changed “Allium Mongolicum” to “Allium mongolicum”, please see the correction in revised version.

(4) L41. Explain whether it is about the smell or the quality of the meat? List the literature cited.

Reply:

Feeding AM could significantly improve quality of mutton, please see the L41-42 in revised version.

(5) L42-43. ‘How to overcome seasonality of AMR?’ Style.

Reply:

‘How to overcome seasonality of AMR?’ has been changed to “How to overcome seasonality is crucial to the utilizing of AM”, please see the L43-44 in revised version.

(6) L60-63. For each example of food, assign literature separately. Example: ‘….fermented rose jams [14], Dezhou braised chicken [15]…’

Reply:

For each example of food, assign literature separately, please see the L63-64 in revised version.

(7) L64. Tricholoma matsutake – Latin names should be written in italics.

Reply:

We have changed “Allium Mongolicum” to “Allium mongolicum”, please see the correction in revised version.

(8) L67.  ‘…yak milk powder were compared by [17] .’ Add the name of the first author of the cited publication.

Reply:

The name of the first author of the cited publication has been added, please see the L68 in revised version.

(9) L73. The sentence ‘Drying is an essential methods to overcome seasonality and extend the storage period of AMR.’ is a repetition of a sentence L50-52.  Rephrase or remove.

Reply:

The sentence ‘Drying is an essential methods to overcome seasonality and extend the storage period of AMR.’ has been removed, please see the L74 in revised version.

(10) L74-L76. ‘However, dried AMR still has not been adopted. Until now, what kind of drying methods can be employed to dry AMR and retain flavour to the maximum extent  possible is still a research gap.’ Use English style!

Reply:

The sentence “However, dried AMR still has not been adopted. Until now, what kind of drying methods can be employed to dry AMR and retain flavour to the maximum extent  possible is still a research gap.” has been changed to “However, dried AM still has not been adopted. Until now, found some kinds of drying methods that can be employed to dry AM and retain flavor to the maximum is still a research gap”, please see the L75-77 in revised version.

(11) Is there a need to dry AMR for human nutrition? I think that AMR is mainly consumed by farm animals (sheep). So, is it not mowed and dried in the sun? I propose to rebuild this paragraph to focus on the effects of drying on the volatile compounds.

Reply:

There are many kinds of the volatile compounds in AMR, fresh AMR has been boiled as a cold dishes in restaurant, and mixed with meat to make steamed stuffed bun in Inner Mongolia. Dried AMR have potential to be used as a spice to be added to stuffing. If AMR was mowed and dried in the sun, the quality of dried AMR is inconsistent due to the drying condition outside is always changed.

We have rebuild the paragraph, please see the L74-82 in revised version.

(12) L80. Change the word ‘utilized’ to ‘used’.

Reply:

We has been changed the word ‘utilized’ to ‘used’, please see L81 in revised version.

(13) L91. ‘The initial moisture content of the sample was 91.93% (w.b.)’. How was it measured?

Reply:

The moisture content was determined by the oven method. Samples were dried in an oven at 105 ℃ until a constant weight was obtained.The moisture content (wet basis) was calculated according to formula (1) as follows:

                                                     (1)

where Wt is the moisture content of the sample (g/g, wet basis); mt is the weight of the sample (g); and mg is the constant weight (g).

(14) L93-94. How was it determined that a sample had a moisture content of less than 8%?

Reply:

Weigh the samples before drying M0, Weigh the samples at any time during drying process Mt. Wt is the moisture content of the sample (g/g, wet basis); when Mt satisfy an equation of , then, drying process end.

(15) L95. At ambient temperature? Enter a value.

Reply:

We have enter the value of ambient temperature 25±5 ℃, please see L96 in revised version.

(16) L118-129. Statistical analysis. How many repetitions was HS-GC IMS analysis performed? How many repetitions of drying were made for each of method?

Reply:

There are 3 repetitions of drying were made for each of method, and no repetitions was HS-GC IMS analysis performed. In order to eliminate test error, the samples used to HS-GC IMS analysis was from homogeneous mixing sample of 3 repetitions of drying.

(17) L97-107. Add drying time to each drying method (FD, VD, HAD).

Reply:

Drying time of each drying method (FD, VD, HAD) has been added, please see the L100-109 in revised version.

(18) L116. ‘…10-100mL/min at 0-20 min…’ Should be 10-20 min?

Reply:

We have been corrected “0-20 min” to “10-20 min”, please see L118 in revised version.

(19) L138-141. Style. Rephrase.

Reply:

We has been rephrased the paragraph, please see L138-143 in revised version.

(20) L152. meant?

Reply:

We has correct “meant” to “mean”, please see L157 in revised version.

(21) L177-179. ‘…the brighter the color, the higher the content.’ What about the red one?

Reply:

The red color indicated the higher content than the bright color. We have explain thoroughly in manuscript, please see L181-183 in revised version.

(22) L272-275. Where do these values come from? In Fig. 5, the axis names and scale are missing.

Reply:

These values come from GC-IMS analyzer equipped with a gallery plot plugin. These values indicated Euclidean distance. The aim of Fig.5 is just to distinguish dying methods, have no axis names and scale. Please see the references below:

(23) L285-287. ‘…increased, decreased and vanished.’ What  does it mean? Is there any trend according to the drying method?

Reply:

We have changed the sentence to “The drying makes the most part of volatile components in fresh AM changed. After drying, the content of some volatile components is increased, the content of some other volatile components is decreased, and some kind of volatile components is vanished”, please see L287-290 in revised version.

Thank you very much for your help again. Please contact us freely if you have questions.

With best regards.

Sincrcely yours,

Zhang Ledao and Wang Guoze

Round 2

Reviewer 1 Report

[Foods] Manuscript ID: foods-1749155

Revised version 2

English still poor making difficult the reading and interpretation. Manuscript needs English editing service

I have just corrected a few errors:

Line 41: Please add the word meat to the sentence or authors can use the following sentence instead: It is found that sheep meat from AM pasture-fed animals tastes better

Lines 42-44. Consider moving the sentence:

However, AM is seasonal and hard to be enjoyed all the year round. How to overcome seasonality is crucial to the utilizing of AM.

to line 51.

Lines 95-97: No need to place the sentence: The dried samples were kept 10 days before testing.

Just add the storage time information as follows: After drying, samples were vacuum-packed and stored for 10 days at ambient temperature (25±5 ℃), for subsequent analyses.

Lines 243- It is sulphide and not sulfide.

Conclusions chapter.

Conclusion should report the major findings.

It is expected to be mentioned here the major differences in volatiles between the drying methods not only in quantity, but more specifically once volatiles were identified and grouped.

Lines 292-294: Remove the sentence: After drying, the content of some volatile components is increased, the content of some other volatile components is decreased, and some kind of volatile components is vanished.

Authors mentioned in the Conclusion chapter… that keep volatile components to the maximum in dried samples was required reason why FD is not suitable to be adopted. If so, they should also place this requirement in the last sentence of the Introduction chapter (line 80).

Reviewer 2 Report

Storage of samples prior to testing (L95-96). What is the reason for such large fluctuations in temperature during storage? It is therefore possible to store one sample at 20 °C and the other at 30 °C. Additionally, changes in temperature can cause ingredients to migrate in the product and release volatile compounds, especially in the case of lyophilization. How long were the samples kept before testing?

L98-101. What was the shelf temperature?

L144-146. How did the authors justify these changes? What could have had a significant impact on this?

L201-202. A discussion is needed. How does drying affect volatile compounds?

L210-212. A discussion is needed. How does freezing affect volatile compounds?

L225-226. A discussion is needed. Explain the effect of the drying method on volatile compounds? What could these differences result from?

L227. A discussion is needed. What can it be related to?

L232-241. Does the identification of volatile compounds in Allium mongolicum Regel coincide between the different determination methods (GC-MS, HS-SPME-GC-MS)? Please compare your results with those of other scientists.

L282-283. What reactions? Expand, give examples.

SI units. Change "hours" to "h" (L100-109). All measurement units we written separately (e.g. 2-10 ml/min, 0.53 mm, -25 °C), except for % (e.g. 8%). Needs improvement.

Round 3

Reviewer 1 Report

Changes were made according to my suggestions, but, again, English must improve.

Author Response

                                                               Jul. 6, 2022

Dear Assigned Editor Serena Wang:

Thank you for your comments concerning our manuscript entitled “The Effects of Drying Methods on Volatile Compounds of Allium mongolicum Regel” (ID: 1749155). The comments are very valuable and helpful for improving our manuscript. We have made modification carefully through an editing service, please see the revised version.

This manuscript has never been submitted for publication in other journals, conference proceedings and the collections of conference papers.

The corresponding address is:

Prof. Guo-Ze Wang                       

School of Life Science and Technology

Inner Mongolia University of Science and Technology, 014010, Baotou, P.R. China

Email: wangguoze@imust.edu.cn

Tel: +86-0472-5951943

Please contact me freely when you have questions.

With best regards,

Sincerely yours,

Le-dao Zhang and Guo-Ze Wang

Reviewer 2 Report

Professional English proofreading is recommended.

Author Response

(The authors gave the same response as above.)
